# Biomarkers for the Evaluation of Immunotherapy in Patients with Cholangiocarcinoma

**DOI:** 10.3390/cancers17030555

**Published:** 2025-02-06

**Authors:** Thaleia-Eleftheria Bousou, Panagiotis Sarantis, Ioanna A. Anastasiou, Eleni-Myrto Trifylli, Dimitris Liapopoulos, Dimitra Korakaki, Evangelos Koustas, Michalis Katsimpoulas, Michalis V. Karamouzis

**Affiliations:** 1University Pathology Clinic, General and Oncology Hospital “Agioi Anargyroi”, National and Kapodistrian University of Athens, Timiou Stavrou 14, 145 64 Kifisia, Greece; thaliaelb@yahoo.com (T.-E.B.); dimitrisliapop@gmail.com (D.L.); dimkorakaki@gmail.com (D.K.); 2Experimental Surgery Unit, Center of Clinical, Experimental Surgery and Translational Research, Βiοmedical Research Foundation of the Academy of Athens, 4 Soranou Ephessiou, 115 27 Athens, Greece; mkatsiboulas@bioacademy.gr; 3Diabetes Center, First Department of Propaedeutic Internal Medicine, Medical School, Laiko General Hospital, National and Kapodistrian University of Athens, Agiou Thoma 17, 115 27 Athens, Greece; anastasiouiwanna@gmail.com; 4Department of Pharmacology, Medical School, National and Kapodistrian University of Athens, Agiou Thoma 17, 115 27 Athens, Greece; 5GI-Liver Unit, 2nd Department of Internal Medicine, National and Kapodistrian University of Athens, General Hospital of Athens “Hippocratio”, 114 Vas Sofias, 115 27 Athens, Greece; elenimyrto.trif@gmail.com; 6Oncology Department, General Hospital Evangelismos, Ipsilantou 45-47, 106 76 Athens, Greece; vang.koustas@gmail.com

**Keywords:** cholangiocarcinoma, immunotherapy, biomarkers, exosomes, interleukins, lymphocytes, immune checkpoint inhibitors

## Abstract

Immunotherapy is now one of the therapeutic options for the treatment of unresectable or metastatic cholangiocarcinoma. Given the poor prognosis of this cancer type and the low response rate to immunotherapy, the application of predictive biomarkers is essential for selecting suitable candidates for treatment with immune checkpoint inhibitors. The aim of this review is to document the biomarkers that have been studied, and have been associated so far, with the effectiveness of immunotherapy in cholangiocarcinoma. Further research on biomarkers from tissue or liquid biopsy, such as characteristics of tumor microenvironment, gene alterations, exosomes, miRNA, and ctDNA, will aid in identifying appropriate patients who will benefit from this treatment and in determining prognosis.

## 1. Introduction

Biliary tract cancer, including cholangiocarcinoma (CCA), gallbladder cancer, and ampullary cancer, is the second most common primary liver cancer and accounts for 3% of all gastrointestinal tumors. Despite the fact that biliary tract cancers are described as a single tumor category, there are several differences between subgroups (intrahepatic CCA, extrahepatic CCA, gallbladder cancer, and ampullary cancer) as far as epidemiology, clinical presentation, gene alterations, treatment options, and prognosis are concerned. Overall, patients with CCA present a poor prognosis with a 5-year survival rate ranging from 2% to 23% [1]. The addition of immunotherapy, especially immune checkpoint inhibitors (ICIs), has shown promising results. However, only a small number of patients are suitable for ICIs using the existing guidelines. Among them, only a limited number have shown sufficient response [2]. As a result, creating a more effective therapeutic protocol (including immunotherapy) based on the tumor’s characteristics and determining useful prognostic and predictive biomarkers is crucial to maximizing the effectiveness of the treatment. Moreover, the difficulty in obtaining sufficient tissue biopsy from patients with CCA, in order to analyze gene mutations and the tumor microenvironment or monitor changes in biomarkers during treatment, shifts the focus of interest to liquid biopsy and the information it can offer. Within the term liquid biopsy, we include every molecule, DNA, and RNA released by the tumor or circulating cancer cells in blood, urine, and bile samples that could be used for early diagnosis, for monitoring changes (genetic, molecular) during treatment, or for the prediction of patient prognosis.

In this review, we will discuss the existing research on the evaluation of immunotherapy in patients with CCA, aiming to highlight the biomarkers which have already studied that could be used as predictive or prognostic factors. Based on the limited data on the use of ICIs in CCA, the tumor micro-environment plays a significant role in tumor development and treatment responsiveness, especially the infiltration by lymphocytes, as well as immune system pathways, including the process of interleukin secretion. Adding to the multiple capabilities that the utilization of the exosomes offers and the wide application of NGS in CCA biopsies, these are three interesting fields (lymphocyte and interleukin levels, exosomes, and gene mutations) that are worth studying further to determine which patients would truly benefit from immunotherapy. The objective of our review is to highlight important literature over the past 15 years, drawing from sources like published research studies, meta-analyses, and some case reports and, specifically, those studies we consider essential to the determination of valuable biomarkers that can predict patient responsiveness to treatment with ICIs. We focus on the importance of searching for promising biomarkers in peripheral blood samples, like exosomes, interleukins, and CD4+/CD8+ ratio, from patients who have received immunotherapy for CCA, as well as on the correlation between gene mutations and the response to ICIs.

## 2. Materials and Methods

We performed a thorough literature search focusing on biomarkers already studied to evaluate immunotherapy in patients with CCA. We queried PubMed for our literature search using the keywords (((“Cholangiocarcinoma”[Mesh]) OR ((cholangiocarcinoma[Title/Abstract]) OR (bile duct cancer[Title/Abstract]))) AND ((“Immunotherapy”[Mesh]) OR (((((immunotherap*[Title/Abstract]) OR (pd-1[Title/Abstract])) OR (pd-l1[Title/Abstract])) OR (pembrolizumab[Title/Abstract])) OR (durvalumab[Title/Abstract])))) AND ((“Biomarkers”[Mesh]) OR ((biomarker*[Title/Abstract]) OR (blood test*[Title/Abstract]))). The data that we reviewed consisted of publications from 2009 to the present, published in English, including primarily completed studies, some clinical trials, and case reports. The literature review was completed in September 2024.

## 3. Cholangiocarcinoma: Features and Therapy

Biliary tract cancers (BTCs) consist of intrahepatic cholangiocarcinoma (iCCA), extrahepatic cholangiocarcinoma (eCCA, further subdivided into perihilar (also called Klatskin tumors) and distal cholangiocarcinoma), gallbladder adenocarcinoma (GBA) and ampullary cancer (located in the ampullae of Vater) [1]. Extrahepatic CCAs are more common than iCCAs, but the incidence of iCCAs has shown a faster increase, which may be due to a more accurate diagnosis compared with the past. CCA originates from the epithelial cells of the bile duct. More than 90% of CCAs are adenocarcinomas divided into three histologic subgroups: mass-forming, periductal-infiltrating, and intraductal-growing [3]. The histopathology of CCA is characterized by the presence of abundant stroma, which infiltrates the tumor environment and contains fibro-genic cells and immune cells [4]. The cellular components of tumor stroma are the tumor-associated macrophages (TAMs), cancer-associated fibroblasts (CAFs), Treg cells, and NK cells. The aforementioned cells stimulate cancer growth, invasion, and recruitment of immune cells by releasing diverse chemokines and growth factors [5]. Tertiary lymphoid structure (TLS) represents a structure which is included in the immune infiltration of TME, containing a B-cell zone and a surrounding T-cell zone with diverse types of T-cells, dendritic cells, and high endothelial venules (HEVs) [6]. Four immune subsets of iCCA have been described based on the composition of the TME. The immune desert phenotype is found in 48% of cases and is characterized by low expression of immune markers. The immunogenic pattern consists of many immune cells and is characterized by the activation of immune checkpoint pathways. The myeloid-rich pattern has a poor prognosis, a moderate to strong expression of myeloid signatures, and a low lymphocytic-high fibroblastic phenotype [7].

Recent evidence suggests that iCCA and eCCA are biologically different cancers with differences in incidence, molecular profile, risk factors, and mortality. Regarding the mutations, *KRAS*, *SMAD4* and *STK11* alterations are more commonly seen in eCCA, whereas mutations in *IDH1*, *BAP1*, *TP53*, and *FGFR2* fusions occur more frequently in iCCAs [8]. Several risk factors are significantly associated with iCCA and/or eCCA, such as choledochal cysts, cholelithiasis, choledocholithiasis, hepatolithiasis, cholangitis, primary sclerosing cholangitis (PSC), biliary cirrhosis, primary biliary cirrhosis (PBC), cholecystitis, cholecystectomy, liver flukes, inflammatory bowel diseases (IBD), duodenal ulcer, chronic pancreatitis, cigarette smoking, alcohol consumption, type II diabetes, obesity, alcoholic or nonalcoholic liver disease, nonspecific cirrhosis, hemochromatosis, and HCV/HBV infection. Cholelithiasis and cholecystectomy primarily affect eCCA, and parenchymal liver diseases are recognized as the main risk factors for iCCA [9].

Patients with eCCA appear with jaundice, dark urine, clay-colored stool, pruritus, hepatomegaly, and mass in the right upper quadrant area. On the other hand, patients with iCCA may appear with correct upper quadrant abdominal pain/tenderness, malaise, night sweats, weight loss, and cachexia. In some cases, paraneoplastic syndromes, like Sweet syndrome, erythema multiforme, and porphyria cutanea tarda, are part of the symptoms. At the same time, other patients are asymptomatic, and CCA is discovered after a differential diagnosis for abnormal liver blood tests [10]. The diagnostic tools for CCA consist of laboratory tests (ALT, AST, γ-GT, ALP, serum bilirubin level, prothrombin time, CEA, Ca19-9), radiologic findings (abdominal ultrasound, CT, MRI-MRCP) and endoscopic findings. Endoscopic retrograde cholangiopancreatography (ERCP) and percutaneous transhepatic cholangiography (PTC) are invasive techniques that are used for diagnostic or therapeutic purposes in patients with bile obstruction. For diagnosing iCCA endoscopic ultrasound, (EUS)–FNA guided ultrasound is more sensitive than ERCP. Other diagnostic tools are intraductal ultrasound (IDUS), which improves the accuracy of the local staging of CCA, and cholangioscopy via peroral or percutaneous transhepatic routes, which can be used for targeted biopsies, lithotripsy, and evaluation of the extent of CCA before surgery. Meanwhile, PET scans and chest CTs are highly recommended for the staging of CCA [10].

The treatment for BTC patients depends on the position of the cancer, the stage at the time of diagnosis, and its molecular profile. For both iCCAs and eCCAs, complete resection followed by adjuvant therapy is the only potentially curative treatment for patients with resectable disease. Based on NCCN guidelines, version 4.2024, primary treatment options for patients with unresectable or metastatic disease include systemic therapy, clinical trial, consideration of locoregional therapy, or best supportive care. Radiation therapy (RT) with concurrent fluoropyrimidine or palliative RT is also an option for patients with unresectable disease. Systemic therapy consists of chemotherapy with or without immunotherapy or radiation therapy and targeted therapy. Regarding chemotherapy, gemcitabine-based or fluoropyrimidine (fluorouracil or capecitabine)-based regimens are the recommended options for advanced BTCs. Particularly, as first line treatment for unresectable or metastatic CCA, the preferred regimens are cisplatin with gemcitabine plus durvalumab or pembrolizumab and, as subsequent lines, regimens like FOLFOX (5-fluorouracil, calcium folinate, oxaliplatin). For tumors MSI-H/dMMR, the recommendation is pembrolizumab, for TMB-H nivolumab plus ipilimumab, and for gene fusion positive tumors, such as NTRK or RET, targeted therapies.

Immunotherapy is included in the therapeutic options for CCA, particularly as part of the recommended therapy as first-line treatment, combined with chemotherapy, for unresectable or metastatic CCA, since it has been proven that the combination of immunotherapy with chemotherapy has significantly improved overall survival. The term “immunotherapy” includes ICIs, cancer vaccines, and adoptive cell transfer. The categories of ICIs are programmed death 1 (PD-1), programmed death ligand 1 (PD-L1), and cytotoxic T-lymphocyte antigen-4 (CTLA-4). The mechanism of action of immunotherapy lies in the reinforcement of the immune system’s ability to recognize cancer cells by overcoming the mechanisms by which tumors suppress the immune response [11]. ICIs overcome the mechanisms that tumors hijack to suppress the antitumor immune response. More specifically, the role of immune checkpoints is to prevent tissue damage when the immune system responds to pathogens, and to preserve tolerance to self-antigens. This is achieved by down-regulating T-cell activation or effector functions. One of the mechanisms by which tumors evade the immune system is the activation of immune checkpoints. As a result, cancer cells and other tumor cells that express immune checkpoints render TILs, particularly CTLs and NK cells, unable to kill them. Durvalumab, the PD-L1 inhibitor, and pembrolizumab, the PD-1 inhibitor, has been approved as a treatment for patients with CCA. According to the TOPAZ-1 trial, durvalumab with a gemcitabine–cisplatin regimen has been indicated as first-line therapy in BTCs, showing improved OS rates [12]. Additionally, according to the KEYNOTE-158 trial, pembrolizumab has been indicated in the treatment of solid tumors with high-frequency microsatellite instability (MSI-H) or high tumor mutational burden (TMB-H) [13]. Thus, pembrolizumab is part of first-line or subsequent-line (for progressive disease, with no prior treatment with an ICI) systemic treatment for unresectable or metastatic MSI-H, dMMR, or TMB-H (for subsequent-line therapy) BTCs [14]. However, MMR is reported in only 1–10% of patients with iCCA [15]. Therefore, based on the guidelines, the only biomarkers used in practice for choosing immunotherapy are MSI-H, dMMR and TMB-H. There is a lack of biomarkers for prediction of responsiveness to treatment and indication of the most appropriate target group, with specific characteristics that will present benefits stemming from the addition of ICIs to the systemic therapy.

## 4. Biomarkers and Cholangiocarcinoma

Several biomarkers have been investigated in relation to early diagnosis, prognosis and therapeutic response in patients with BTC, mainly those who have not been treated with immunotherapy. In this context, we will discuss specific biomarkers that might provide valuable insights into predicting response treatment if examined in patients undergoing immunotherapy. These are divided into non-invasive biomarkers and biomarkers in tumor tissue. Non-invasive biomarkers are further divided into circulating tumor DNA (ctDNA), microRNA, cytokines, metabolites, extracellular vesicles (EVs), and circulating tumor cells (CTCs) [16]. In CCA, considering that taking enough tissue biopsy for molecular testing is challenging, liquid biopsy has been highlighted as a useful diagnostic tool and a contributing factor to therapeutic decision-making and prognosis prediction. Firstly, ctDNA is easily detected and allows the identification of genomic alterations in patients with CCA, resulting in the selection of targeted therapy and the prediction of prognosis. MicroRNAs associated with the development of CCA and with potential use as biomarkers are as follows: miR-21, miR-221, miR-222, miR-192, miR-483-5p, miR-150, miR-26a, miR-1281, miR-126, and miR30b. CA19-9 and carcinoembryonic antigen (CEA) constitute the most widely used protein biomarkers for diagnosing and monitoring CCA, but they are characterized by low sensitivity and specificity. Cytokines with potential diagnostic value and/or predictive value are as follows: CYFRA 21-1, MMP7, osteo-pontin, peri-ostin, mesothelin, IL-6, S100A6, DKK1, SSP411, and suPAR, TGF-β1. Circulating metabolites that may be helpful in the assessment of early diagnosis, prognosis, or treatment monitoring are glycol-cholic acid, glycol-chenodeoxycholic acid, androsterone sulfate II, dehydroepiandrosterone, LysoPC(14:0), (15:0), 21-deoxycortisol, bilirubin, ChoE(22:6), (20:4), CMH(d18:1/16:0), PC(16:0/16:0), PC(34:3), histidine, bile acids, cholesterol/lipid, phosphatidy-choline, SM(42:3), (43:2), PC(0–16:0/20:3), (0–18:0/18:2), SM(d18:2/16:0), and Cer(d18:1/16:0) [16,17]. Additionally, many diagnostic, prognostic and predictive biomarkers are studied based on EVs selected from diverse biological samples. For instance, EVs and their cargoes (proteins, miRNA, piRNA) have a promising potential in differentiating CCA from HCC or benign conditions, and in early diagnosis of CCA in patients with PSC [17,18]. Another valuable tool for clinical decision-making among those targeted therapies that also have prognostic value is provided by CTCs. Particularly, CTC count has been associated with poor prognosis [17]. As far as tumor tissue biomarkers are concerned, specific genetic alterations, non-coding RNAs, the presence of MSI-H, and the percentage of CD8+ PD-1high are some of the findings that are linked with more accurate diagnosis, therapeutic option, and prognostic prediction [19]. For example, mismatch repair deficiency (MMR) indicates a good response to antiPD-1/PD-L1 monoclonal antibodies. A1-Antitrypsin deficiency is more common among patients with CCA and has been linked to longer survival, indicating that CCA tumors in these patients may be less aggressive [17].

Moreover, claudin-18.2 (a member of the tight junction protein family) upregulation might be useful in distinguishing benign from malignant bile duct lesions, improving cytological diagnosis, while correlating with poor survival and lymph node metastasis in iCCA [20]. However, further studies must be carried out to show novel biomarkers and determine the effectiveness of immunotherapy in the field of CCA treatment.

## 5. Biomarkers Associated with the Immunotherapy Landscape

Many studies regarding response and resistance in ICI treatment for solid tumors have shown valuable results that can be applied in selecting patients who will benefit from immunotherapy as part of their therapeutic plan. However, the data are limited to that for immunotherapy in patients with CCA. Here, we present some promising biomarkers studied in other solid tumors that need further study in CCA to highlight their association with this type of cancer.

Hereby, we present some experimental tissue biomarkers, which open up future perspectives. It has been reported that Anaplastic Lymphoma Receptor Tyrosine Kinase ALK has a correlation with anti-PD-1 response [21]. POLE/POLD1 mutations are associated with improved OS in cancer patients treated with ICIs. Additionally, Gene expression-based (GEB) signatures could show off immunological transcriptomic phenotypes that predict the activity and toxicity of immunotherapy. Interferon-γ (IFN-γ), a significant immune system cytokine, has been studied and evaluated, combined with related gene signatures. Other experimental tissue biomarkers include single-cell RNA sequencing (scRNA-seq) and HLA I expression. DNA damage repair (DDR) gene mutations, such as in poly (ADP–ribose) polymerase 1 and 2 (PARP) or breast-related cancer antigens (BRCA), may lead to antitumor immunity, making possible the use of (DDR) gene mutations as a predictive biomarker for immunotherapy response [22].

Moreover, circulating biomarkers, such as circulating CD4+ and CD8+ T-cell ratio, levels of circulating myeloid-derived suppressor cells (MDSCs) or Treg, soluble forms of PD-1 (sPD-1) and PD-L1 (sPD-L1), serum sCTLA-4 levels, ctDNA and CTCs, cytokines including IFN-γ, IL-6, IL-8, IL-11, and IL-2, VEGF levels and high pre-treatment plasma levels of angiopoietin-2, neutrophil-to-lymphocyte ratio (NLR) along with or without LDH, and *Bacteroides*-enriched and *Akkermansia*-enriched gastrointestinal microbiota, are all potential options with promising prognostic impact [23,24]. All these results come from studies that include participants treated with ICIs for melanoma, NSCLC, breast, colorectal, or urothelial cancer. Thus, it is vital to conduct new research to study immunotherapy’s effectiveness, especially in CCA, and highlight biomarkers with applications in therapeutic decisions and response monitoring [25].

## 6. Biomarkers for the Evaluation of Immunotherapy in Patients with CCA

### 6.1. Tissue Biopsy Biomarkers

Most of the studies regarding the identification of novel biomarkers for the use of ICIs in CCA are based on tissue biopsies and further examination with diverse methods, such as immunohistochemistry (IHC), in situ hybridization (ISH), and next-generation sequencing (NGS). Specifically, they have focused on TME and its differentiation among immunotherapy patients. As previously mentioned, there is a complex interplay between tumor cells and immune cells in the desmoplastic microenvironment in CCA. A more detailed understanding of this interaction may open new therapeutic options with maximum benefit for the patient.

Three biomarkers are clinically validated to predict ICI response: PD-L1, TMB, and MSI/dMMR, with the last two also having prognostic value [24,26]. The study of Yang et al. included 139 patients treated at two major hospitals in China and explored the prognostic value of MSI-H and PD-L1 expression for overall survival (OS) and progression-free survival (PFS). The findings indicated that patients with MSI-H tumors experienced longer OS and PFS compared to those with microsatellite stable (MSS) tumors when treated with PD-1 inhibitors. Additionally, integrating MSI-H status with PD-L1 expression provided further prognostic insights, with patients showing a PD-L1 combined positive score (CPS) ≥5 having better OS and PFS outcomes. However, the role of PD-L1 as a biomarker remains contentious, as other studies have shown inconsistent results regarding its predictive value in BTC [27]. MSI-H status was associated with higher numbers of tumor-infiltrating immune cells and immune checkpoint molecule expression in a CCA cohort. Specifically, correlation analysis of MSI with tumor-infiltrating immune cells, MHC I, and PD-L1 expression showed a high number of CD8+ T-cells, FOXP3+ regulatory T-cells, CD20+ B cells, and high MHC I expression in MSI-H CCAs, suggesting that these parameters could be correlated with better OS in CCA treated with immunotherapy [28].

A meta-analysis investigated the role of PD-L1 expression as a predictive biomarker in patients with BTC undergoing anti-PD-1/PD-L1 therapy. The study found that, although there were no significant differences in the objective response rate (ORR) and disease control rate (DCR) between PD-L1 positive (+) and negative (−) patients, those with PD-L1 (+) tumors had longer progression-free survival (PFS) and overall survival (OS) This suggests that PD-L1 expression is more valuable as a biomarker for predicting survival than treatment response [26]. It is worth mentioning that an increased PD-L1 expression after one cycle of chemotherapy presents an improved PFS based on the study of Zeng et al., suggesting a therapeutic-responsive PD-L1 expression, rather than baseline PD-L1 levels, as a promising biomarker for better prognosis [29].

Job et al. classified 198 iCCA into four subtypes based on the tumor microenvironment (TME), each associated with distinct immune escape mechanisms and patient outcomes. One of these subtypes, characterized by an inflamed TME, could potentially be treated with checkpoint blockade immunotherapy [7]. Moreover, Fontugne et al. analyzed 58 iCCA and 41 eCCA cases, finding that PD-L1 was mainly expressed by intra-tumoral immune cells, principally in tumors with dense intra-tumoral lymphocytic infiltration. As a result, patients with CCA with dense intra-tumoral lymphocytic infiltration could be good candidates for PD-L1/PD-1 inhibitors [30,31]. A small trial (ChiCTR2000036652) of 12 patients indicated that the gene expression profile of pre-treatment tumor biopsies might predict clinical outcomes to combined immune–chemotherapy in iCCA patients, emphasizing the importance of a tumor microenvironment characterized by active T-cell migration with regards to a better outcome for the treatment [29]. Chen et al. conducted a multi-omic analysis of 16 iCCA patients, and divided the tumor samples into two groups, one high-immune and one low-immune group. They revealed that higher densities of CD8+ T-cells, CD4+ T-cells, and CD20+ B cells were infiltrated in tumor samples from patients in the first group. Moreover, they found an upregulation of immune pathways in high-immune tumors, such as the T-cell receptor signaling pathway and cytokine–cytokine receptor interaction pathway, and a higher expression of the IFN signature, which is predictive of anti-PD-1 treatment response in diverse solid tumors. These results imply that tumors in the high-immune group may have a positive response to anti-PD-1 therapy [32].

Cytotoxic T-cell (CTL) markers and TIDE can be useful in classifying CCA tumors as CTL-high or CTL-low, allowing clinicians to better predict the likelihood of response to immunotherapy. For CTL-high CCA tumors, TIDE can assess if T-cell dysfunction is likely to hinder response while, for CTL-low tumors, it can evaluate whether ICB may enhance immune cell infiltration. Okawa et al. examined 219 patients with BTC, finding significant correlations between PD-1 expression and tumor location, sex, and T-cell expression, with higher PD-1 levels in eCCA, males, and T-cell-high expressed tumors [33]. On the contrary, iCCAs and GBCs seem to have increased levels of PD-L1 expression, TMB-H, and MSI-H compared to eCCA, based on Weinberg et al. [34]. Additionally, Jung et al. found that patients with more infiltrated T-cells (cytotoxic and memory T-cells) close to the tumor benefited from ICIs [35]. Kida et al. found that CCA patients with two or more tumor-associated antigen (TAA)-specific cytotoxic T lymphocyte responses had significantly prolonged overall survival, suggesting that such patients may benefit more from immunotherapy. High lymphocyte counts in peripheral blood also correlated with better TAA-specific responses. As a result, patients with high lymphocyte counts may benefit more from immunotherapy [36]. Zhou et al. analyzed the expression of several immune markers (CD8, CD4, Foxp3, and PD-L1) using immunohistochemistry (IHC) in 11 tissue biopsies from patients with CCA receiving anlotinib and toripalimab. NGS sequencing showed that *STK11* gene mutation was more frequent in the group with poor outcome, while patients with a higher CD8/Foxp3 ratio had significantly longer survival [37].

Shang et al. investigated the role of tertiary lymphoid structures (TLS) in predicting the response to ICIs in a cohort of 100 patients. They found that, while the expression of a four-gene TLS signature was not associated with tumor cell PD-L1 expression or OS, a high TLS score in pre-treatment tumor tissues was associated with better outcomes in patients receiving immunotherapy [38].

Zhou et al. studied 95 iCCA patients and found that high endothelial venules (HEVs) could categorize patients into low-HEV and high-HEV subtypes. The high-HEV subtype showed upregulation of several immune checkpoints, suggesting a more robust immune response [39].

Tan et al. selected 47 eCCA tissue blocks in the University Hospital RWTH, Aachen. They showed that patients with high NFD (large numbers of small nerve fibers in the TME, not invaded by cancer cells) are associated with higher PD-1 expression. This indicates that NFD might serve as a prognostic biomarker for these patients [40].

Tsuchikawa et al. reported the association between epithelial–mesenchymal transition (EMT) related factors and PD-L1 expression in the eCCA and its role as a biomarker for patient prognosis. A whole-exome sequencing study of BTC showed that a subgroup with hypermutations, such as PIK3CA, presented high expression of immune checkpoint molecules, including PD-L1. They suggested that a subgroup of tumors with an EMT phenotype might be a potential target for treatment using ICIs [41,42]. Moreover, the findings of Cao et al. imply that CCA with mesenchymal and cancer stem cell (CSC) traits could potentially be treated using ICIs, providing a new avenue for targeted therapy in these patients [43].

Another outcome of the study of Cao et al. was that the immune checkpoints, including IDO1, PD-L1, FASLG, CD80, HAVCR2, CD73, CTLA-4, LGALS9, VTCN1, and TNFRSF14, are associated with poor prognosis in CCA patients. Combining PD-1, PD-L1, and CTLA-4 with modulators, such as TIGIT, TNFRSF14, and FASLG, further highlighted poor outcomes. These findings suggest that these immune modulators could be used as prognostic biomarkers in CCA patients [43].

T-cell immunoreceptor with immunoglobulin and ITIM domain (TIGIT) is a novel checkpoint inhibitory receptor expressed on immune cells, while lymphocyte-activation gene 3 (LAG-3) is an immune checkpoint receptor protein on the surface of effector T-cells and regulatory T-cells (Tregs). Both receptors represent a promising perspective in immuno-oncology with interesting studies over recent years. Tang et al. studied LAG-3, an immune checkpoint protein co-expressed with PD-1. In advanced BTC, LAG-3 expression correlated with higher objective response rates, greater tumor shrinkage and longer progression-free and overall survival. Enriched CD8+ T-cells in LAG-3-positive tumors indicate their role as a biomarker for immune-inflamed tumors and chemoimmunotherapy sensitivity [44]. Heij et al. studied immune checkpoints, like PD-1, PD-L1, PD-L2, TIM3, LAG3, and ICOS, in iCCA, with CD8_PD-L2 and CD4_ICOS_TIGIT linked to nodal metastases and poor prognosis [45]. More research must be conducted in order to extract clearer results, with application in clinical practice.

### 6.2. Biomarkers Based on Gene Mutations

Gene alterations are another category of biomarker that has been studied to predict better outcome in patients with CCA receiving ICIs. The genomic alterations of iCCA and eCCA are revealed using either tumor tissue or cfDNA and offer important information on novel therapies and the profile of patients who would benefit from ICI treatment.

Tumor mutation burden (TMB) is a significant genomic biomarker that predicts the response to ICIs, like PD-(L)1 and CTLA-4 inhibitors. TMB is calculated by counting the total number of somatic mutations identified through exome sequencing of the coding region of the genome. Studies have demonstrated that tumors with high TMB are more likely to respond positively to ICIs. For instance, a pan-tumor analysis of 12 clinical trials involving pembrolizumab (including 1772 patients) showed that patients with TMB-H (defined as ≥175 mutations per exome) had significantly better outcomes when treated with pembrolizumab. Moreover, TMB-H, especially when combined with high microsatellite instability (MSI-H), is a strong independent predictor of positive responses to anti-PD-(L)1 therapy, regardless of PD-L1 status. Respectively, the predictive value of TMB-H is even more pronounced when coupled with PD-L1 expression. Overall, the findings suggest that TMB, mainly when defined with higher thresholds (such as 16 mutations per mega-base using the TSO500 assay), can serve as a critical biomarker for predicting the efficacy of anti-PD-L1 therapy in a wide range of tumors, including CCA [35].

A study involving 98 patients with advanced biliary tract cancer (BTC), 34 of whom were treated with camrelizumab (an immune checkpoint inhibitor) combined with GEMOX (a chemotherapy regimen), explored the impact of genomic alterations on prognosis and immunotherapy outcomes. Researchers identified 11 key mutated genes (e.g., *APC*, *ARID1A*, *ERBB2*, *LRP1B*, *TNFAIP3*) that played significant prognostic roles. They found that the wild-type (non-mutated) version of these genes responded better to camrelizumab compared to the mutated version. The study also utilized the TIDE algorithm to predict tumor responses to ICIs in a cohort from The Cancer Genome Atlas (TCGA). TIDE is a biomarker database and algorithm set used to predict immunotherapy response by modeling T-cell dysfunction and tumor exclusion. This prediction model showed that the wild-type subset had a superior objective response rate (ORR) compared to the mutated subset. Combining genomic classification with an evaluation of the TME further enhanced the stratification of immunotherapy outcomes, particularly in patients with the wild-type gene signature. The study also discovered that co-mutations in the *KRAS* and *TP53* genes in advanced CCA were associated with favorable responses to immunotherapy. In contrast, tumors with only a single *KRAS* mutation had poorer prognoses and less favorable outcomes from immunotherapy, underscoring the importance of specific genetic contexts in treatment efficacy [46].

Guo et al. showed that the 8-IRDEGs signature, which includes *RORA*, *CNTFR*, *COLEC10*, *TNFSF15*, *SRC*, *PDGFD*, *TUBB3*, and *PLXNB3*, plays a crucial role in tumor immunity. This gene signature can effectively represent immune cell infiltration in CCA patients. Its application can assist clinicians in selecting patients who would benefit from immunotherapy. Moreover, they showed the superiority of 8-IRDEGs compared with the TIDE algorithm as a predictive factor [47].

*IDH1* mutations are associated with immunosuppression in the tumor environment. Inhibiting IDH1 may activate the immune system against cancer cells by converting an immune “cold” environment into a “hot” one, which is more likely to respond more favorably to ICI treatment. Further studies should be conducted on the correlation of IDH1 mutations and the response to immunotherapy [48]. An ongoing clinical trial of treatment with a combination of IDH1 inhibitors and durvalumab is worth mentioning in patients with unresectable or metastatic solid organ tumors (NCT04056910) [49].

Research by Mody et al. explored the relationship between PD-L1 expression and various genomic alterations in BTC. The study found significant associations between PD-L1 expression and mutations in genes like *BRAF*, *BRCA2*, *RNF43*, and *TP53*. These associations suggest that the recognition of these gene alterations could have a role in the selection of patients for treatment with ICIs, and that PD-L1 expression could help identify promising biomarkers for more targeted treatment approaches [50].

20 iCCA and 20 adjacent tissue samples were obtained from patients who underwent resection of iCCAs between 2016 and 2019 at the Affiliated Cancer Hospital of Zhengzhou University. Focusing on specific immunosuppressive genes of tumor-associated macrophages (TAMs), Xu et al. indicated that, of the selective 10 tumor-promoting genes of TAMs, only *MMP19* and *SIRPα* could predict ICI response in iCCA patients [51].

According to Ji Shin et al., the DNA checkpoint mutated, defined as genomic alterations in seven genes, i.e., checkpoint kinase 1 (*CHEK1*), checkpoint kinase 2 (*CHEK2*), DNA repair-associated (*BRCA1*), the serine/threonine kinase ATM, the serine/threonine kinase ATR, mediator of DNA damage checkpoint 1 (*MDC1*) and tumor protein p53 binding protein 1 (*TP53BP1*), which may serve as useful biomarkers for predicting the effectiveness of ICI therapy [52]. Wang et al. identified a set of 54 genes through a model which pinpointed a specific subtype, S2, characterized by high expression of immune checkpoint genes, making it a good candidate for immunotherapy. This result highlights the significance of the molecular phenotype of the tumor and, specifically, the expression of genes involved in regulating the immune system’s response against cancer cells [53].

In the study by Sui et al., it was reported that two patients with iCCA, who exhibited a high rate of insertion-deletion mutations (indels), responded positively to a combination of PD-1 blockers and chemotherapy. Note that indels and single-nucleotide variants (SNVs) determine the TMB together. This finding suggests that a high ratio of indels could be a potential new predictor for how well iCCA patients will respond to PD-1 blockade therapy [54].

Chida et al. study showed that the IFNγ pathway was upregulated in patients who responded to ICI treatment, consistent with previous studies that linked an IFNγ-related mRNA profile to better outcomes with ICIs across various cancers. Gene signatures detected via RNA sequencing, particularly those induced by IFNγ, may serve as potential predictive biomarkers for the effectiveness of ICIs in microsatellite-stable (MSS) and mismatch repair-proficient (pMMR) solid tumors [55].

Another study conducted across three cohorts, including 187 patients with hepatobiliary cancers, used copy number variations (CNVs) in plasma cell-free DNA (cfDNA) to construct a risk score model for predicting survival in patients receiving ICI-based therapy. They showed that patients with lower CNV risk scores had significantly longer overall survival (OS) and progression-free survival (PFS) than those with high CNV risk scores. This suggests that CNV risk scores could serve as a valuable tool for predicting immunotherapy outcomes [56].

Li et al. developed a patient-specific circulating tumor DNA (ctDNA) fingerprint panel based on NGS, targeting high-frequency clonal populations within the tumors. This new ctDNA platform proved effective in monitoring treatment responses across various cancer types, including CCA, offering a promising tool for tracking therapeutic effectiveness, especially in patients undergoing multiple treatments [57]. A case report from Yu et al. demonstrates the potential utility of tumor-informed ctDNA in CCA as a response monitoring tool that can guide therapy optimization. The report showed that early intervention with immunotherapy may lead to improved survival outcomes [58].

### 6.3. Circulating Biomarkers Such as Proteins, circRNA, miRNA

Apart from gene mutations, circRNA, miRNA, proteins, and immune cell expression levels, epigenetic alterations are promising tools that could significantly predict response to immunotherapy in patients with CCA. Many of the studies referred to below are based on the correlation between some specific features of patients with a good outcome receiving ICI and the possible modification of indications for ICI treatment for CCA.

Li et al. showed that circSLCO1B3 (circRNA, which originates from exon 9 to exon 15 of the *SLCO1B3* gene) increases PD-L1 protein expression by inhibiting the ubiquitin–proteasome pathway, promoting immune evasion in iCCA. This finding suggests that circSLCO1B3 could be a valuable biomarker for patient selection in immunotherapy [59].

A study by Lee et al. highlights the potential of circRNA as a novel diagnostic biomarker for identifying iCCA patients who may benefit from PD-1 blockade therapies. Their research suggests that pembrolizumab can rejuvenate exhausted T-cells, making certain iCCA types with high levels of T-cell exhaustion good candidates for immunotherapy. The study also identified PTPN22 and circ-ADAMTS6 in plasma exosomes, which are associated with a subgroup of immunogenic Asian iCCA characterized by T-cell exhaustion and neutrophil extracellular traps (NETs). This suggests that plasma exosomal circRNAs could help detect this specific iCCA subgroup, which may respond well to immune checkpoint blockade [60]. As previously mentioned, exosomes can be used as prognostic and predictive factors for CCA. The study of Trifylli et al. also highlighted the application of exosomes in evaluating immunotherapy in CCA. Specifically, bile EV-MiR-183-5p increases PD-L1 expression and, as a result, could predict the immune tolerance of iCCA [18].

Peng et al. used the TIMER database to analyze the relevance of MIR155HG and the currently available blocking molecules with superior therapeutic effects, i.e., PD-1, PD-L1, CTLA4, LAG3, and TIM3. A median or higher positive correlation between MIR155HG and immune checkpoint molecules suggests that MIR155HG could be a valuable predictor of immunotherapy effectiveness [61].

Pan-cancer analysis of CMTM6 protein expression showed that PD-L1 expression was positively correlated with CMTM6 expression in CCA, implying that high CMTM6 expression could respond better to anti-PD-1/PD-L1 immunotherapy [62].

Zhao et al. identified Cul3 as a suppressor of CCA in vivo. CRISPR-Cas9 screening in knockout mice found that Cul3 deficiency raises Nrf2 and Cyclin D1 levels, promoting CCA development. scRNA-Seq showed reduced cytotoxic T-cell activity in tumors lacking Cul3. Testing anti-PD1/PD-L1 therapy revealed an effective response, which was enhanced by combining it with sorafenib [63].

In a study by Sun et al., five cohorts of iCCA patients were analyzed in order to understand the impact of immunotherapy. Single-cell data from 12 iCCA biopsies, including paired samples taken before and after immune checkpoint blockade (ICB) therapy, revealed a significant upregulation of CD73 expression in malignanT-cells following treatment. This suggests that CD73 could potentially be utilized as a biomarker for predicting a patient’s response to ICB therapy, potentially guiding treatment decisions [64].

Wu et al. investigated the expression of CCDC6 and its correlation with immune cell infiltration in CCA using the TIMER and TIMER 2.0 databases. Despite the fact that they did not find a strong link between CCDC6 expression and the types of immune cells infiltrating CCA, they did observe a weak association between CCDC6 and key immune checkpoint molecules, like PD-1, PDCD1, and CTLA4, in CCA from the GEPIA database. These findings suggest that CCDC6 might have the potential to be a predictive biomarker during targeted and immunotherapy treatments [65].

Moreover, high dermatopontin (DPT) levels were associated with enhanced responses to anti-PD-1/PD-L1 inhibitors in CCA. The study uncovered that DPT secreted by CCA cells could stimulate macrophages to secrete the chemokine CCL19, which may explain how DPT enhances immune cell infiltration within the TME [66].

Yang et al. analyzed data from 147 iCCA patients treated with anti-PD-1 therapy at Sun Yat-sen University. They discovered that higher levels of serum lipids correlated with better overall survival (OS). Notably, in multivariate analysis, APOA1 and triglycerides (TGs) were identified as independent predictors of OS. The study developed a prognostic nomogram incorporating four factors: CA19-9, APOA1, tumor number, and TG. This nomogram demonstrated superior predictive value for 1-year and 2-year survival compared to using serum lipids alone [67].

A case report on a young woman with eCCA showed that, despite lacking standard predictive features such as PD-L1 expression, the patient responded well to pembrolizumab, potentially due to high HLA class I and II antigen expression. The report concluded that determining HLA class I and II antigen expression could be helpful in patients potentially eligible for immune checkpoint therapy [68].

### 6.4. Other Biomarkers

A study by Qiu et al. identified 3369 common differentially methylated regions (DMRs) in 105 patients with BTC. BTCs without many methylation changes were infiltrated with CD8+ lymphocytes, and had high PD-L1 expression, indicating an inflamed TME, potentially with a better immunotherapy efficacy [69].

In a cohort of 698 patients with metastatic BTC, 39 patients were identified with Epstein–Barr virus-associated intrahepatic cholangiocarcinoma (EBVaICC). Among 205 patients who received PD-1 antibody therapy, those with EBVaICC showed significantly longer overall survival (OS) than those with EBV-negative ICC, highlighting the potential impact of viral status on treatment outcomes [70] (Table 1, Figure 1).

## 7. Adverse Events of Immunotherapy and Associated Predictive Factors

Despite the encouraging results from the addition of immunotherapy to therapeutic protocols for various types of cancer, including CCA, immunotherapy sometimes triggers the activation of the immune system towards different organs, causing immune-related adverse events (irAEs). The National Cancer Institute’s Common Terminology Criteria for Adverse Events (CTCAE) has categorized the toxicities into five widely applicable grades. Severe to life-threatening irAEs (grade ≥ 3) occur in 20–30% of patients treated with anti-CTLA-4 agents, 10–15% of patients with PD-1 inhibitors, and 55% of patients receiving anti-CTLA-4/PD-1 combination therapy. In the phase III CheckMate 067 trial of nivolumab plus ipilimumab versus ipilimumab or nivolumab for advanced melanoma treatment, irAEs of any grade occurred in 96% of patients receiving combination therapy and 86% of those treated with monotherapy, and grade 3 or 4 AEs occurred in 59%, 28%, 21%, respectively [71]. Systems that may be affected are the cardiovascular, the respiratory, the endocrinological, the gastrointestinal, the neurological, the urinary, or the skin. According to the NCCN guidelines, version 1.2024, patients may develop myocarditis, pneumonitis, hypophysitis, hypothyroidism, primary adrenal insufficiency, colitis, pancreatitis, meningitis/encephalitis, peripheral neuropathy, myasthenia gravis, Guillain Barre syndrome, vision changes, arthritis, renal failure, dermatitis, Steven Johnson syndrome, or sicca syndrome. Interestingly, many studies have related the development of certain irAEs to an improved response to ICIs. Therefore, it is crucial to research biomarkers that will predict the possibility of side effects. Except for some risk factors that have been associated with the occurrence of irAEs, such as age, sex, and history of an autoimmune disease, many biomarkers have been recorded for predicting these side effects [72]. These biomarkers could be categorized into cytokines, serum proteins, circulating blood cells, autoantibodies, HLA genotypes, and microRNAs. IL-6, IL-8, IL-10, TNF-α, albumin levels, levels of each white blood cells (WBC) subgroup, and platelet number are some of the biomarkers that have been studied and could have clinical applications in therapeutic decisions [73]. A promising and useful research field consists of showcasing predictive biomarkers for irAE, considering the significant impact among patients receiving ICIs and the further association of irAE with the response to immunotherapy, especially in CCA.

## 8. Limitations

There are numerous limitations regarding the application of the results of the aforementioned studies in clinical practice. Firstly, the sample size is limited, not only due to the rarity of the disease but also because of the not widely implemented use of immunotherapy in cholangiocarcinoma. As a result, some studies include various types of cancer, such as hepatocellular carcinoma or others, or refer to biomarkers generally related to cholangiocarcinoma rather than specifically to immunotherapy treatment. Some results have been derived from case reports that include only one or two patients. Τhe practical difficulties of obtaining histological biopsies and the need for monitoring during treatments make it necessary to promote non-invasive techniques [74]. Additionally, due to the poor prognosis of cholangiocarcinoma, long-term monitoring of changes in various biomarkers that could be used to predict the response to this treatment is not feasible. Another limitation is the challenging application of biomarkers, such as exosomes, in everyday practice. On the other hand, biomarkers that are often used, like cytokines, have low sensitivity and specificity [16]. Finally, as already noted, intrahepatic and extrahepatic CCA differ in terms of pathophysiology, molecular analysis, and prognosis. However, studies that distinguish the degree of response to immunotherapy based on tumor location and the frequency of various biomarkers in each type are lacking.

## 9. Conclusions

BTC includes various types of cancer that are distinguished based on location, phenotype, treatment, and prognosis. The poor prognosis of BTC, combined with its increasing incidence and limited therapeutic options, highlights the urgent need for new treatments and biomarkers that can predict prognosis and help select appropriate patients for each therapy. Obtaining sufficient biopsy material from patients with CCA is often challenging, and discovering blood-based biomarkers is essential. Biomarkers from diverse biological samples, like blood, urine, and bile, have already shown promising results. Among the existing biomarkers that have been studied, the type of circulating lymphocytes has generated significant interest; however, further study is needed to highlight statistically significant results. Additionally, they represent a promising biomarker with potential for broad application in routine practice, unlike other biomarkers that, although they have demonstrated impressive results, are challenging to apply, such as exosomes. Nonetheless, further study of exosomes related to immune therapy response in patients with CCA is warranted. Furthermore, biomarkers associated with the response to immunotherapy studied in other cancer types, such as various interleukins, could potentially also provide helpful information for CCA patients. Their measurement through flow cytometry, a widely used method, appears promising for correlating interleukins with the prognosis of patients receiving immunotherapy. Finally, considering that NGS sequencing is now widely used on CCA biopsies for therapeutic decision-making, the correlation of specific gene mutations, such as *IDH1*, with the response to immunotherapy represents an interesting area of research.

## Figures and Tables

**Figure 1 cancers-17-00555-f001:**
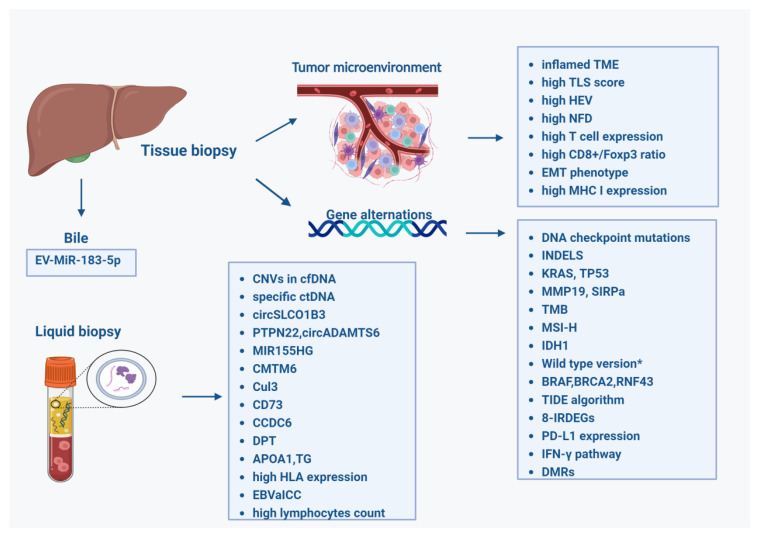
Predictive biomarkers for patients taking ICIs for CCA. (Created in BioRender.com). * not-mutated version based on Xiaofeng Chen et al. study of 11 key mutated genes (e.g., APC, ARID1A, ERBB2, LRP1B, TNFAIP3).

**Table 1 cancers-17-00555-t001:** Biomarkers in patients with CCA and their correlation with the response to immunotherapy.

Biomarkers	Type/Source	Correlation with Outcome	Patients Number	Reference
TLS	TME	A high TLS score in pre-treatment tumor tissues was associated with better outcome	100	[38]
HEV	TME	High-HEV subtype showed upregulation of immune checkpoints, suggesting a better response	95	[39]
NFD	TME	High NFD is associated with a higher PD-1 expression, indicating that NFD might be a prognostic biomarker	47	[40]
TME-inflamed	TME	High-immune group have an upregulation of immune pathways and a positive response to anti-PD-1 therapy	374	[7,29,31,32]
TMB	genomic	TMB-H tumors (defined as ≥175 mutations per exome) had better outcomes with pembrolizumab	125	[35]
MSI	genomic	MSI-H tumors had longer OS and PFS compared to those with MSS tumors when treated with PD-1 inhibitors	139	[27,28]
*MMP19*, *SIRPa*	genomic	Among genes of TAMs, only *MMP19* and *SIRPα* could predict ICI response in iCCA patients	20	[51]
EMT	genomic	EMT phenotype might be a potential target for treatment using ICIs	117	[41,43]
CNV	genomic	Patients with lower CNV in cfDNA risk scores had longer OS and PFS with ICI-based treatment	187	[56]
INDELS	genomic	Tumors with a high rate of indels responded positively to a combination of PD-1 blockers and chemotherapy	2	[54]
Gene signature	genomic	Wild-type (non-mutated) version of 11 key mutated genes (e.g., *APC*, *ARID1A*, *ERBB2*, *LRP1B*, *TNFAIP3*) responded better to camrelizumab	98	[46,47,48,50]
		Gene alterations of *BRAF*, *BRCA2*, *RNF43*, and *TP53* could have a role in the selection of patients who will be treated with ICIs	652	
		TIDE algorithm showed that the wild-type subset had a superior objective response rate (ORR) compared to the mutated subset	98	
		Co-mutations in the *KRAS* and *TP53* genes in CCA were associated with favorable responses to immunotherapy	98	
		8-IRDEGs signature can represent immune cell infiltration in CCA patients, predicting response to immunotherapy	330	
		*IDH1* mutations are associated with immunosuppression in the TME. Inhibiting *IDH1* makes more possible a better response to ICI	14	
DNA checkpoint mutations	genomic	DNA checkpoint mutated, defined as genomic alterations in seven genes, may predict the effectiveness of ICI	62	[52,53]
*STK11*	genomic	*STK11* gene mutation was more frequent in the group with poor outcomes treated with anlotinib and toripalimab	11	[37]
PD-L1 expression	tissue	PD-L1 expression is more valuable as a biomarker for predicting survival rather than treatment response Therapeutic-responded PD-L1 expression, rather than the baseline PD-L1 levels correlated with better PFS Integrating MSI-H status with PD-L1 expression provided prognostic insights, with patients showing CPS ≥ 5 having better OS and PFS	887	[26,27,29]
CMTM6	protein	High CMTM6 expression could respond better to anti-PD-1/PD-L1 immunotherapy	database	[62]
CD73	protein	Upregulation of CD73 expression in malignanT-cells following treatment, suggesting that CD73 could predict the response to ICI	12	[64]
CCDC6	protein	Weak association between CCDC6 and immune checkpoint molecules like PD-1, PDCD1, and CTLA4, suggesting that CCDC6 might be a predictive biomarker during immunotherapy	99 and database	[65]
DPT	protein	DPT secreted by CCA cells stimulates macrophages to secrete the chemokine CCL19 and enhances immune cell infiltration in TME and response to ICIs	database	[66]
IFN-γ	genomic	IFNγ pathway was upregulated in patients who responded to ICI treatment	2	[55]
Serum lipids	metabolic	APOA1 and TG were independent predictors of OS. A nomogram of CA19-9, APOA1, tumor number, and TG demonstrated superior predictive value	147	[67]
CircSLCO1B3	cirRNA	CircSLCO1B3 increases PD-L1 expression, promoting immune evasion in iCCA, suggesting that it could be a valuable biomarker for patient selection for immunotherapy	55	[59]
PTPN22, circ-ADAMTS6	cirRNA	PTPN22 and circ-ADAMTS6 in plasma exosomes are associated with a subgroup of immunogenic ICC, suggesting that plasma exosomal circRNAs could help detect the ICC subgroup, which may respond well to ICIs	14	[60]
MIR155HG	Non-coding RNA	Positive correlation between MIR155HG and immune checkpoint molecules suggests that MIR155HG could be a useful predictor of immunotherapy effectiveness	10	[61]
CD8+ cells, CTLs	TME	Patients with more infiltrated T-cells (cytotoxic and memory T-cells) close to the tumor benefited from the ICIs	219	[33,35,36]
CD8+/FOXP3	TME	Higher CD8/Foxp3 ratio had significantly longer survival	11	[37]
ctDNA	Liquid biopsy	Patient-specific ctDNA fingerprint panel proved effective in monitoring treatment responses	27	[57,58]
EBVaiCCA	Viral	EBVaICC showed significantly longer OS compared to those with EBV-negative ICC	39	[70]
DMR	epigenetic	BTCs without many methylation changes were infiltrated with CD8+; they had high PD-L1 expression, indicating an inflamed TME, potentially with a better immunotherapy efficacy	105	[69]
HLA class	genomic	High HLA class I and II antigen expression could indicate patients eligible for ICIs	302	[68]
Cul3	protein in laboratory animals	Cul3 deficiency reduced cytotoxic T-cell activity. Testing anti-PD1/PD-L1 therapy in these tumors revealed an effective response	Animal experiment	[63]
EV-MiR-183-5p	Bile	Bile EV-MiR-183-5p increases PD-L1 expression and, as a result, predicts the immune tolerance of iCCA	-	[18]
MHC I	genomic	High MHC I expression in MSI-H CCAs has been correlated with better OS in CCA patients treated with immunotherapy	308	[28]
lymphocytes	blood cells	Patients with high lymphocyte counts may benefit more from immunotherapy	26	[36]

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
