# Peer review of "Biomarkers for the Evaluation of Immunotherapy in Patients with Cholangiocarcinoma"

_cancers, 2025, doi:10.3390/cancers17030555_

Round 1
Reviewer 1 Report
Comments and Suggestions for Authors
The manuscript presented by Bousou et al. aims to review the available literature in terms of biomarkers with potential use to identify responders to immunotherapy within cholangiocarcinoma patients.
The topic is relevant, however, the manuscript must be improved, since in its present form is not suitable for publication.
First, the presentation of the manuscript is poor.
Examples: references are inserted in no consistent form, sometimes the period is placed before the reference and sometimes is placed after the refence. Sometimes the period is missing. This is seen numerous times. Lines 61, 64, 116, 119, 223 and so on.
Authors refer gene alternations, did they mean alterations. Lines 33, 59, 225, 366, 419 and so on.
Authors introduced several abbreviations such as CCA, iCCA, eCCA, however later in the manuscript they used once again cholangiocarcinoma instead. Please use the terms consistently. Line 285, 509
Line 106 it seems repetitive, extrahepatic CCAs and eCCA sara the same.
Line 131, replace primarly by primary
Line 159 a brief description of the NCCN guidelines would be helpful
Line 196 rewrite, it is confusing.
Line 219 add reference for the information presented.
Gene names for human should be presented in capitol letters and italicized.
Bacteroides and Akkermansia should be italicized
Line 292 re write, there being..
When authors refer to other authors, sometimes the name is italicized and sometimes not. Present authors name consistently.
CTLA-4 is another relevant immune checkpoint and it was not included in the search, as described in methods.
Line 415 a reference for this trial is missing.
Line 424 should read associated.
In table 1, sentences should start with capitol letters (section correlation with outcome).
Lines 550-553, there was not description of the grades 0-3.
Line 567 what is WBC group
Conclusions section is too much. Conclusions must be concise.
Comments on the Quality of English Language
A few improvements would be helpful. Some of these are pointed in my report.
Reviewer 2 Report
Comments and Suggestions for Authors
This manuscript attempts to review what is known about potential biomarkers in patients with cholangiocarcinoma which could be used to guide further management with immunotherapy.
The way the information is summarized in the manuscript is such that it is difficult for the reader to appreciate just how limited the evidence base is for most of the biomarkers which have been investigated. There are other review articles which have been published in recent years which have presented similar information in a far clearer manner for eg publications from Macias RIR et al one of which is as follows- https://ajp.amjpathol.org/article/S0002-9440(24)00277-3/fulltext
Hence the information in Table 1 needs to be reformatted so that it is far more readily apparent as to how many patients were in each study for each of the relevant biomarkers.
Other issues include-
1) More attention needs to be provided to what biomarkers are currently been used clinically and why so many of the biomarkers which have been investigated to date are not. The fact that there are limitations to obtaining the relevant tissue at times from the tumor (which is mentioned both in the Introduction and Discussion sections), needs to be supported by referencing the relevant literature.
2) Mention needs to be made as to precisely which of the Guidelines recommend immunotherapy (as per the Introduction section), as well as whether as yet there is any Guideline consensus for use of any of the Biomarkers mentioned in this review
3) What was the cutoff date for the literature review (this is not stated in the Methods section)?
Reviewer 3 Report
Comments and Suggestions for Authors
Thaleia-Eleftheria Bousou et al. perform a very nice review on CCC and immunotherapy. This is a hot topic. the work is very well structured and keeps the reader interested.
The M&M are solid and the results of the review are good. I would like to see a picture of a histological evaluation of some of these markers since markets such as MSI and PD-L1 are evaluated by histology. It would provide some visual interest to the paper, if possible.
Still on histology, the (possible) role of TIGIT and LAG3 should have some extra description since it has been studied in the last years. There are some papers regarding it (https://doi.org/10.3390/onco4030010, PMID: 32716559, PMID: 35656508, PMID: 35464264, PMID: 39751894)
Also in this field one or two sentences on claudin 18.2 would raise some discussion points (PMID: 39731204, PMID: 34486479)
Round 2
Reviewer 2 Report
Comments and Suggestions for Authors
The manuscript appears to have been adequately revised in response to the points made by the reviewer